# Development of Fast Protection System with Xilinx ZYNQ SoC for RAON Heavy-Ion Accelerator

**Seung-Hee Nam [1] , Changwook Son [2] and Jungbae Bahng [3,4,*]**

1   POSTECH Pohang Accelerator Laboratory, Pohang 37673, Republic of Korea; namsh@postech.ac.kr
2   Heavy-Ion Accelerator Research Institute, Daejeon 34000, Republic of Korea; scwook@ibs.re.kr
3   Department of Radiation Oncology, Kangwon National University Hospital,
    Chuncheon 24289, Republic of Korea
4   Department of Accelerator Science, Sejong Campus, Korea University, Sejong 30019, Republic of Korea
*   Correspondence: bahngjb@knuh.or.kr

**Abstract:** The development of the fast protection system (*FPS*) was driven by the critical need to safeguard internal components of the accelerator from beam damage and minimize operational downtime. During accelerator operation, various faults can occur, posing a significant risk. The *FPS* acts as a rapid response system, initiating a shutdown signal to a reliable chopper system to prevent beam damage and ensure the operational availability of the accelerator. To meet the stringent shut off time requirements specific to critical faults, the *FPS* was designed to respond within 50 μs, while the total *FPS* time, including acquisition, redundancy, and processing, needed to be less than 20 μs. In order to achieve these goals, a customized *FPS* was developed for the RAON heavy-ion accelerator, utilizing the Xilinx ZYNQ system-on-chip (*SoC*). The *FPS* system comprised seven acquisition modules, one mitigation module with an embedded *SoC*, and employed optical fiber connections for efficient data transmission. This article provides a comprehensive account of the design, development, and testing of the *FPS* system. Experimental tests were conducted to validate its performance. These tests included verifying the accuracy of cyclic redundancy checks, acquiring interlock signals in short pulses, and measuring the delay time during abnormal signal occurrences. Of particular significance is the measurement of the total signal processing time for a 1 km optical cable in the RAON system, which was determined to be 9.8 μs. This result successfully met the stringent requirement of 20 μs for the *FPS* time. The ability of the *FPS* to operate within the desired time frame demonstrates its effectiveness in protecting the accelerator's components from beam damage and minimizing downtime. Consequently, the *FPS* ensures the operational availability of the accelerator while maintaining the safety and integrity of its internal systems. By providing a detailed account of the *FPS*'s design, development, and testing, this article contributes valuable insights into the capabilities of the *FPS* in real-world accelerator scenarios. The successful implementation of the RAON-optimized *FPS* with the Xilinx ZYNQ *SoC* reaffirms its effectiveness as a fast and reliable protection system, thus enhancing the overall operational performance of the accelerator.

**Keywords:** fast protection system; accelerator control system; Xilinx ZYNQ; experimental physics and industrial control system; RAON heavy-ion accelerator

## 1. Introduction

The Rare Isotope Science Project (*RISP*), conducted by the Institute for Basic Science (*IBS*) in Korea, aims to develop a rare isotope accelerator facility with high beam energy and power output capabilities [1]. The accelerator consists of various subsystems, including control, radio frequency, injector, vacuum, magnet, and cryogenic systems. Among these subsystems, the Machine Protection System (*MPS*) plays a crucial role in safeguarding the internal components of the accelerator from beam-induced damage. As the beam power increases, the need for swift accelerator shutdown to mitigate potential damage

becomes imperative. To achieve this, a transition from a programmable logic controller (*PLC*)-based interlock system to a field-programmable gate array (*FPGA*)-based interlock system is necessary. The *MPS* is responsible for detecting abnormal signals from local devices within the accelerator, such as low-level radio frequency, beam diagnostics, magnet power supplies, and cryogenic systems [2]. When a trip signal is detected, the *MPS* activates the beam stop device, consisting of components like the low-energy beam transfer, chopper, radio frequency quadrupole, and low-level radio frequency, to rapidly halt the beam. The *MPS* is divided into two main systems based on signal acquisition time: the fast protection system (*FPS*) [3] and the slow interlock system (*SIS*) [4]. The *FPS*, which has attracted significant attention as accelerator technology advances, focuses on achieving high equipment availability and stability. To improve availability and stability, the *FPS* must be carefully designed to meet the specific requirements of the diverse devices within the accelerator.

The *MPS* is responsible for detecting abnormal signals from local devices within the accelerator, such as low-level radio frequency, beam diagnostics, magnet power supplies, and cryogenic systems. When a trip signal is detected, the *MPS* activates the beam stop device, consisting of components like the low-energy beam transfer, chopper, radio frequency quadrupole, and low-level radio frequency, to rapidly halt the beam. The *MPS* is divided into two main systems based on signal acquisition time: the fast protection system (*FPS*) and the slow interlock system (*SIS*). The *FPS*, which has gained significant attention as accelerator technology advances, focuses on achieving high equipment availability and stability. To improve availability and stability, the *FPS* must be carefully designed to meet the specific requirements of the diverse devices within the accelerator [5]. In heavy-ion accelerators, a device yield stress may occur when the beam collides with the accelerator main device [6], depending on the type of ion beam mass, kinetic energy, and angle of incidence.

The design and functionality of the *FPS* are critical in ensuring its effectiveness as a fast and reliable beam protection system. Therefore, it is crucial to set clear requirements for the *FPS* prior to its production and assess its practicality by benchmarking against desired outcomes. In heavy-ion accelerators, the occurrence of yield stress in accelerator components due to beam collisions depends on factors such as the ion beam mass, kinetic energy, and incident angle [6]. The heavier the ion weight, the greater the stopping power, and the higher the angle of incidence, the greater the effect [7], where the stopping power of an ion beam is calculated using the Stopping and Range of Ions in Matter (*SRIM*) code. Previous studies at the Spallation Neutron Source (*SNS*) and the Facility for Rare Isotope Beams (*FRIB*) have investigated the impact of beam energy and incident angle on yield stress time, providing valuable insights [8,9].

The time required for the *FPS* to operate effectively includes abnormal beam signal detection, *FPS* processing, and mitigation device operation. By analyzing abnormal beam operation scenarios for various ions, including uranium and oxygen, as well as the melting time of accelerator components like cavities and vacuum chambers, the *FPS* requirements can be accurately determined [10]. In most cases, a response time of approximately 50 μs is sufficient to prevent beam damage on the beam line components, considering the incident beam's grazing angle. Moreover, the melting time of beam line components made of materials like SUS304 for vacuum chambers or niobium for superconducting cavities has been calculated as ~46 μs and ~69 μs, respectively, when it is hit by a uranium beam energy of 18.9 MeV/u in a right angle, which is the fastest loss case, providing crucial insights into protection time requirements.

The *FPS* and *SIS* operate at microsecond and a millisecond scales, respectively. Abnormal signals from most of the collected devices go through the *SIS* to reach the *FPS*, and some of the other collected devices directly send their abnormal signals to *FPS*. Upon receiving abnormal signals, the *MPS* must fully execute the entire process within 50 μs to protect the accelerator from sudden beam losses by alerting the mitigation devices, such as *LEBT* chopper and *RFQ* amp to shut down the beam. Equations (1)–(3) show the calculation

of beam shut-off latency using the parameters used for the equation, as presented in Table 1.

$$T_{Diag} + T_{FPS} + T_{MD} \leq 50 \ \mu sec, \tag{1}$$

$$T_{Diag} = (T_{BPM} \ or \ T_{BLM} \ or \ T_{BCM}) + T_{DFN}, \tag{2}$$

$$T_{FPS} = \sum_{n=1}^{7} (T_{L_n N_n} + T_{N_n N_{n-1}}) + T_{MN} \tag{3}$$

**Table 1.** Parameters used to calculate beam shut-off latency.

| Parameters | Description |
|---|---|
| $T_{Diag}$ | Beam loss diagnosis time (within 25 μs) |
| $T_{FPS}$ | *FPS* process time (within 20 μs) |
| $T_{MD}$ | Mitigation device process time (within 5 μs) |
| $T_{BPM}$ | Beam position monitor fault logic |
| $T_{BLM}$ | Beam loss monitor fault logic |
| $T_{BCM}$ | Beam current monitor fault logic |
| $T_{DFN}$ | Fault signal transition time |
| $T_{L_n N_n}$ | Link node processing time |
| $T_{N_n N_{n-1}}$ | Transmission time between nodes |
| $T_{MN}$ | Mitigation node processing time |

In summary, the *FPS* and *SIS* play vital roles in protecting the accelerator from sudden beam losses. The *FPS* operates at the microsecond scale and ensures swift response times for abnormal signals, while the *SIS* operates at the millisecond scale, facilitating the transmission of abnormal signals to the *FPS*. To effectively protect the accelerator, the entire *MPS* process, including signal detection and mitigation device activation, must be completed within a specified time frame. In this article, we present a detailed account of the design, development, and testing of the *RAON*-optimized custom *FPS*, utilizing the Xilinx ZYNQ system-on-chip (*SoC*), in order to meet the fast shut-off requirement within 20 μs.

## 2. RAON Central Control System

The *RAON* central control system has a middleware to combine multiple experimental devices into one united control system. It uses *EPICS* middleware to integrate local control systems into *RAON* central control systems. The *RAON* central control system has a three-tier architecture. Figure 1 shows a simplified architecture of the *RAON* central control system. The *EPICS* middleware framework is used to integrate various systems into one unified system. The unified central control system, installed in a central control room, supervises every component of the accelerator by monitoring all signals from distributed local devices. Basically, by integrating different systems into one, the central control system classifies all possible derived systems by its functionality and standardizes them. By doing so, it assures safety, reliability, scalability, and availability.

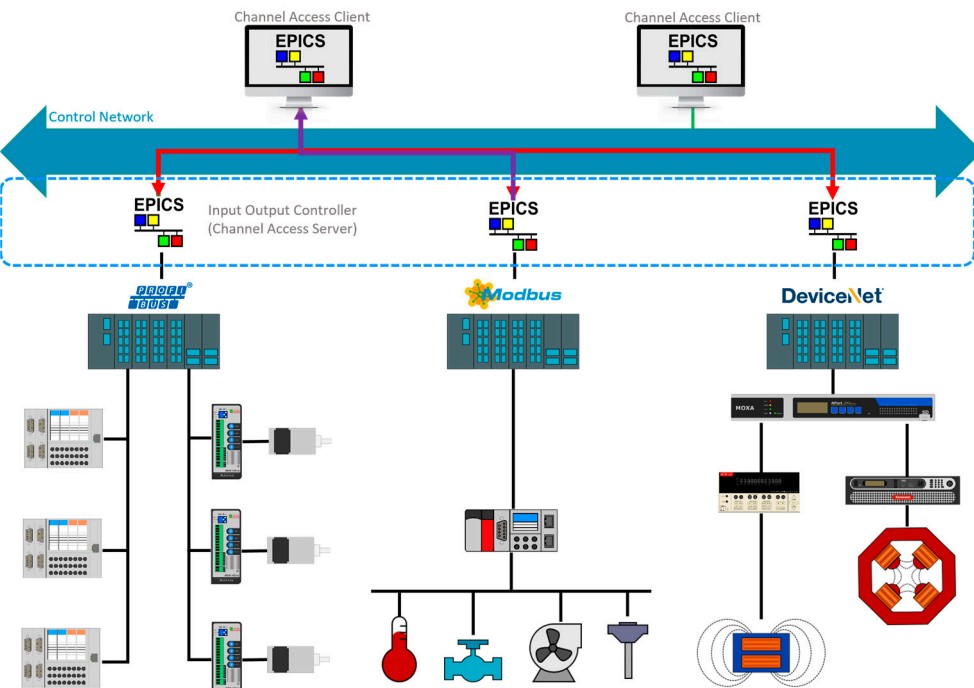

**Figure 1.** *RAON* central control system architecture.

## 3. Composition of Fast Protection System

The *FPS* mainly protects the internal components of the accelerator from beam damage. Prior to designing the *FPS*, an investigation was performed on the *FPS* of major accelerator facilities. The *FPS* consists of two distinct parts, namely, *FPGA* and a central processing unit (*CPU*), which play a role in parameter management, fast protection, fault data, and fault analysis. Figure 2 shows the block diagram that protects equipment in various modes of operation, integrating subsystem interfaces, software development using PLCs, hardware configuration and control interlocks, and *FPGA* control-based timing systems.

1. Mitigation node
2. Acquisition node

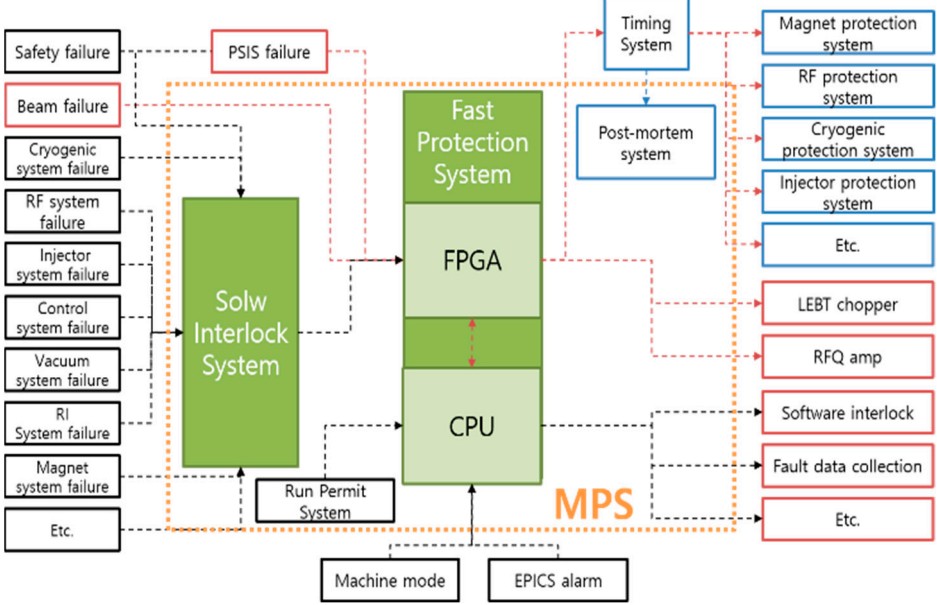

**Figure 2.** RAON machine protection system architecture.

Basically, the *FPS* consists of a mitigation (master) node and an acquisition (slave) node. A proper firmware configuration design of the mitigation and acquisition nodes is required to enable the nodes to transmit and store fault data for fault analysis.

### 3.1. Mitigation Node

The mitigation node consists of a main board for signal processing and data storage and a daughter board for *FPS* device control. The main board has a memory and buffer arranged around the multiprocessor system-on-chip (*MPSoC*). The front panel consists of eight SFP+ transceiver modules, Ethernet transceivers, SD cards, and USB to JTAG ports, and the rear panel consists of connectors for data output to the daughter board. Table 2 presents the specification of the mitigation node prototype.

**Table 2.** Specification of the mitigation node.

| Item | Specification |
|---|---|
| Chassis | Power supply, RF board, digital board, I/O port for external connection, LCD, and LEDs for status<br>Chassis (19-inch 2U-height rack)<br>Power: Single-phase 220 V, noise shielding<br>Cooling system for power |
| *MPSoC* | *MPSoC*: ZYNQ Ultrascale + XCZU9EG-2FFVB1156I |
| SFP+ | I/O port: 8 ea |
| Output | Output port: 24 ea<br>Specification of output: +5 VDC<br>Connection method: hard wiring connection, SMA type |
| Interface | Serial console UART port<br>JTAG port<br>Ethernet port<br>SD card |
| EVG | Embedded *EVR* code receive |

The *PL* of the mitigation node receives the status information of the *FPS* external device input from the acquisition node at 1 μs interval to the postmortem system (*PMS*) data module through the GTH interface. If an abnormal signal is detected, the cutoff signal is transmitted to bthe output of 24 channels, and then status information is stored in the BRAM inside the PL, and a log file is created [11]. In addition, the time stamp and event code received from the event generator (*EVG*) are delivered to the $^{EVR}$ module via the GTH interface. The time stamp data is stored in BRAM along with external device status information stored in real time. BRAM stores not only the status signals of external devices but also built-in test (BIT) data, such as firmware version, temperature data, and status information of the mitigation node [12].

PetaLinux is ported in the *PS* area. In addition, the *EPICS IOC* operates the *FPS* system. The interface with the *PL* is connected to the AXI Interconnect using the device driver inside the kernel to control BRAM and external devices. The *EPICS IOC* uses processing values (PVs) to create the elements that are needed for system operation and data transfer. *OPI* is used to control and monitor the generated *PV* [13]. Figure 3 shows the *PL* and *PS* configurations of the mitigation node.

### 3.2. Acquisition Node

The acquisition node consists of a main board for signal processing and data storage and a daughter board for external device input. The main board has a memory and buffer arranged around the *SoC*. The front panel contains four SFP+ transceiver modules, an Ethernet transceiver, an SD card, and a USB-to-JTAG port. The rear panel consists of a

connector to receive data from the daughter board. Table 3 presents the specification of the acquisition node prototype.

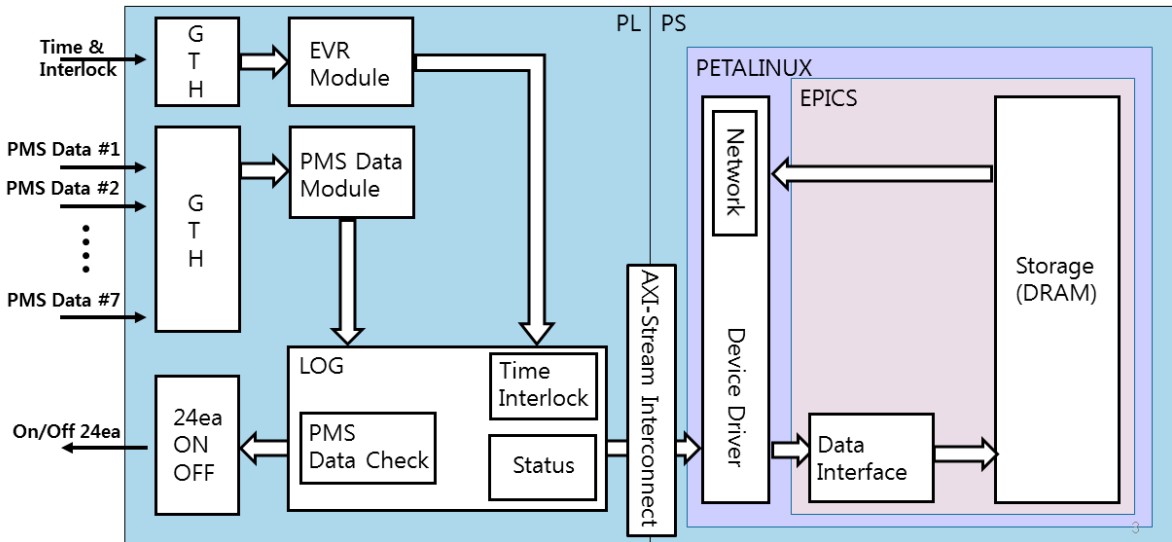

**Figure 3.** Configuration of the mitigation node firmware.

**Table 3.** Specification of the acquisition node.

| Item | Requirement |
|---|---|
| Chassis | Power supply, RF board, digital board, I/O port for external connection, LCD, and LEDs for status<br>Chassis (19-inch 2U-height rack)<br>Power: Single-phase 220 V, noise shielding |
| MPSoC | *SoC*: ZYNQ XC7Z100-2FFG900I |
| SFP+ | I/O port: 4 ea |
| Output | Input port: 64 ea<br>Specification of input: 32 ea dry contacts, 32 ea wet contacts<br>Connection method: Terminal Block type |
| Interface | Serial console UART port<br>JTAG port<br>Ethernet port<br>SD card |
| *EVG* | Embedded *EVR* code receive |

The acquisition node *PL* area receives 64 input data in the *FPS* external device. This data is stored in BRAM along with the time stamp data received from the *EVR*. Data stored in BRAM is transferred to the upper acquisition node or mitigation node. When the *PMS* event code is received from the *EVG*, the reception time is transmitted to the *PS*. PetaLinux is ported in the *PS* area. The *EPICS IOC* operates the *FPS*. The *PS* uses AXI Interconnect to control the interworking with the *PL* and external devices. The data stored in BRAM in the *PL* area is continuously stored in the DDR memory. When the *PMS* event code is received from *EVG*, $\pm 200$ μs data are generated in the form of a file with a time stamp and stored in SDRAM based on the time of occurrence. The *EPICS IOC* generates and transmits BIT and other necessary elements in a *PV* form, and *OPI* controls and monitors the *PV*. Figure 4 shows the firmware configuration of the acquisition node.

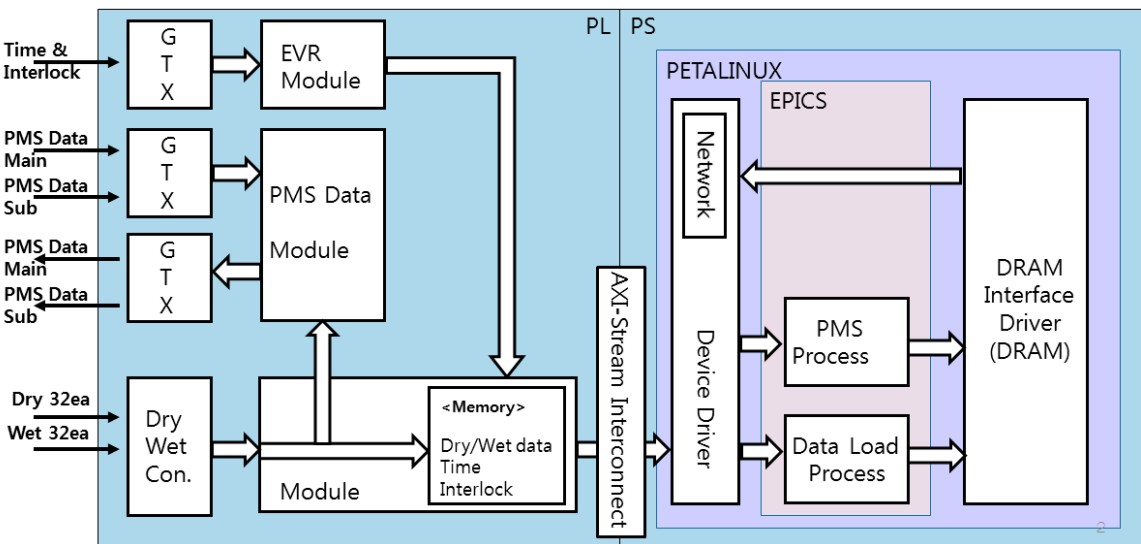

**Figure 4.** Configuration of the acquisition node firmware.

## 4. Development of FPS with Xilinx ZYNQ for Minimizing a Latency

In order to reduce the latency for quick shut off, the device drive software and the minimization of the distance between the two gates in *FPGA* were optimized. As shown in Figure 5, two mitigation nodes and seven acquisition daisy chains can be found in each of the *FPS* primary products. One of the two mitigation nodes acts as a backup in case of failure. Each acquisition chain is composed of seven acquisition nodes that are connected linearly. One mitigation node and one acquisition chain are randomly selected to be tested. With the test result, the remaining one mitigation node and six acquisitions chains are modified for improvement. In the case of the mitigation node, the *SoC* chip was replaced with XC7Z100-2FFG900I for bidirectional communication with the acquisition node.

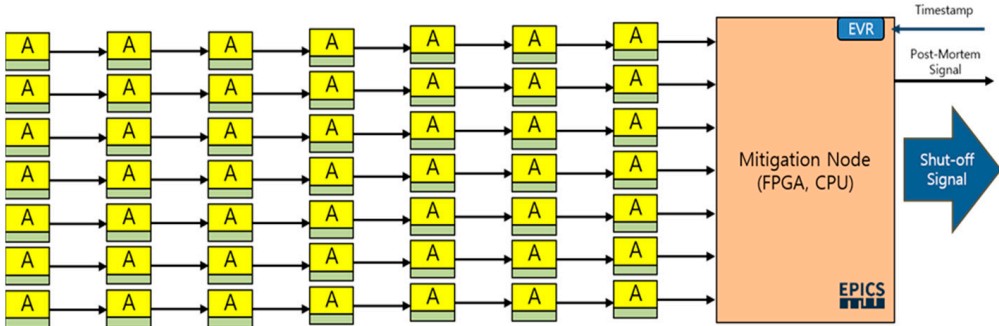

**Figure 5.** *FPS* layout.

Minimizing latency is a crucial aspect of developing a highly efficient Fast Protection System (*FPS*). This article presents a descriptive analysis of a study conducted to optimize latency reduction in *FPS* using Xilinx ZYNQ, a powerful embedded OS porting system. This research aims to enhance device drive software and *FPGA* gate distance for quick shut off. Xilinx ZYNQ emerges as a powerful solution for minimizing latency in the *FPS*. Its versatile functionality, adaptability to changing requirements, real-time processing capability, reduced power consumption, and independent power rails enable the development of a highly efficient and flexible *FPS* control system. By leveraging the advantages and features of Xilinx ZYNQ, developers can enhance the device drive software and *FPGA* gate distance, resulting in significant latency reduction and improved performance in the *FPS*. ZYNQ XC7Z100-2FFG900I [14] and ZYNQ XCZU9EG-2FFVB1156I [15] are implemented for RAON fast protection system.

A spare of the front SFP+ port is created and modified, giving a total of nine SFP+ ports. A masking function is added to prevent a BPS mode setting or input/output of a false trigger signal. In the case of the acquisition node, a spare SFP+ port was also created, giving a total of five SFP+ ports. To receive the BPM output signals directly, the converter terminals were modified to 16 SMA-type input ports. A software change was made to check the BIT of communication status, temperature, time, and timing through LED, LCD, and *OPI* on the node front. The IP and configuration files for each node were automatically updated at each node boot. Additionally, a node self-test mode was added via a software developed.

### 4.1. Modified Mitigation Node

The modified mitigation node is a 2U-high 19-inch rack module. The interior of the module consists of one PCB with *FPGA*, one power supply, cables, and chassis. The following are the changes in the modified mitigation node.

1.  SFP+ port on the main board (9 EA to 10 EA)
2.  *MPSoC* to ZYNQ XC7Z100
3.  Ethernet PHY (Marvell to TI)
4.  Dip-type connector for booting device selection
5.  Wet contact driver in the daughter board
6.  Change in the SMA library
7.  Front SFP+ port (8 EA to 9 EA)

Table 4 presents the specification of the modified mitigation node.

**Table 4.** Hardware specification of the modified mitigation node.

| Classification | Requirement | Implementation Detail |
|---|---|---|
| Configuration | Chassis | 19-inch 2U-height rack<br>Front panel: SFP+ port 9 ea, RJ-45 1 ea, USB to JTAG 1 ea, DSUB-9 1 ea,<br>SD card 1 ea, LED and character LCD for display status<br>Rear panel: SMA 24 ea for output, 220 V AC power<br>Cooling system: fan for cooling |
| | Digital board | SoC: Xilinx ZYNQ XC7Z100-2FFG900I<br>Memory: DDR 1 GB SPI Flash 16 MB<br>Temperature sensor<br>Transceiver, buffer, etc. |
| | SMPs and cable | Power and signal connection flame-retardant cable |
| Interface | SFP+ (9) | *EVG* 3.25 Gbps speed |
| | Ethernet (1) | 1 Gbps |
| | USB (1) | USB to JTAG |
| | UART to JTAG (1)<br>Output (24) | 115,200 bps UART<br>Wet contact (SMA) |

Figure 6 shows the configuration diagram of the modified mitigation node.

The modified mitigation node consists of a main board for signal processing and data storage and a daughter board for outputting signals to the *FPS* devices. The main board is equipped with ZYNQ for PetaLinux, memory, buffer, and sensor, where PetaLinux is ported. The front interface consists of nine SFP+ transceiver modules, Ethernet transceivers, SD cards, and USB to JTAG port. The rear panel is composed of a connector for outputting data to the daughter board. The daughter board then converts output signal that is received from the main board to wet signal and transmits the signal to an external device through an SMA connector.

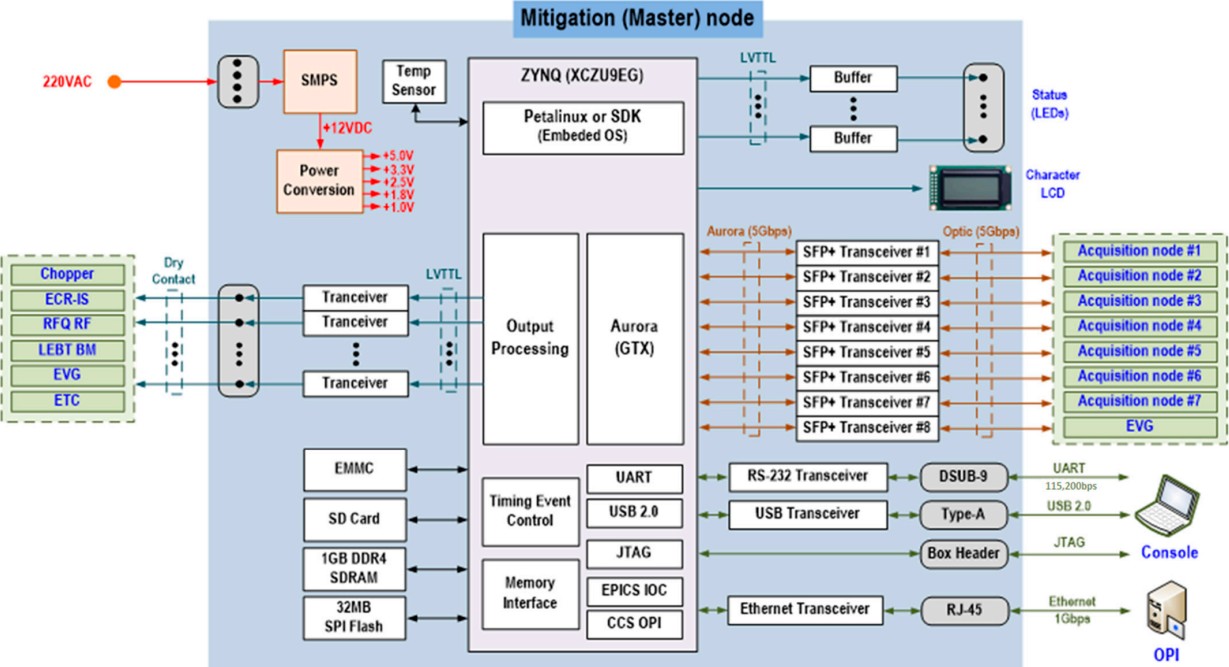

**Figure 6.** Configuration diagram of the modified mitigation node.

### 4.2. Modified Acquisition Node

The modified acquisition node is a 2U-high 19-inch rack module. The module consists of a PCB with *FPGA* and a power supply, cables, and chassis to supply power. The following are the changes in the modified acquisition node.

1. Ethernet PHY (Marvell to TI)
2. Dip-type connector for booting device selection
3. Replaced dry contact 16 EA to SMA
4. Modified SMA driver
5. SMA library

Table 5 presents the product specification of the modified acquisition node.
Figure 7 shows the configuration diagram of the modified acquisition node.

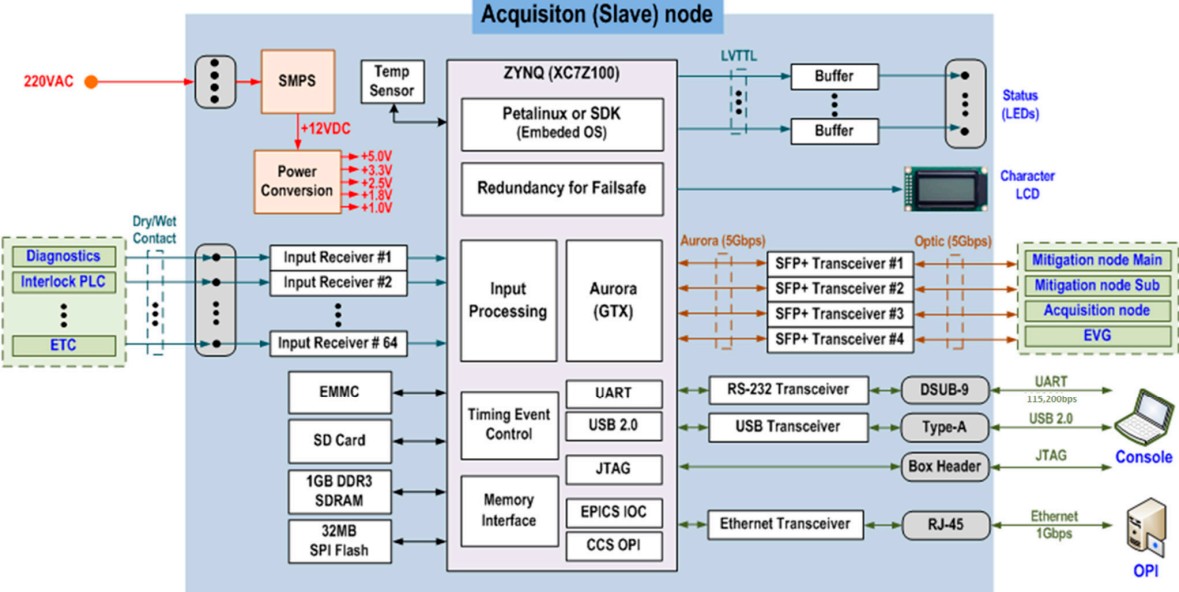

**Figure 7.** Configuration diagram of the modified acquisition node.

**Table 5.** Hardware specification of the modified acquisition node.

| Classification | Requirement | Implementation Detail |
|---|---|---|
| Configuration | Chassis | 19-inch 2U-height rack |
| | | Front panel: SFP+ port 5 ea, RJ-45 1 ea, USB to JTAG 1 ea, DSUB-9 1 ea, SD card 1 ea, LED and character LCD for display status |
| | | Rear panel: terminal block 48 ea for output, SMA 16 ea, 220 V AC power |
| | | Cooling system: fan for cooling |
| | Digital Board | SoC: Xilinx ZYNQ XC7Z100-2FFG900I |
| | | Memory: DDR 1 GB SPI Flash 16 MB Temperature sensor |
| | | Transceiver, buffer, etc. |
| | SMPs and cable | Power and signal connection flame-retardant cable |
| Interface | SFP+ (5) | EVG 3.25 Gbps Speed Single mode |
| | Ethernet (1) | 1 Gbps |
| | USB (1) | USB to JTAG |
| | UART to JTAG (1) Output (64) | 115,200 bps UART Dry contact 32 ea, wet contact 32 ea |

The modified acquisition node consists of a main board for signal processing and data storage and a daughter board for receiving input signals from the *FPS* devices. The main board is equipped with a memory, buffer, and sensor mainly on ZYNQ, where PetaLinux is ported. The front interface consists of five SFP+ transceiver modules, an Ethernet transceiver, an SD card, and a USB to JTAG port, and the rear panel consists of a connector for outputting data to the daughter board. The daughter board is equipped with a terminal block and an SMA to receive 32 dry contacts and 32 wet contacts from the external *FPS* devices. The signal received from the external device is converted into low-level TTL (LVTTL) signal and transmitted to the main board.

## 5. Testing of the FPS

Tests were performed on the time of the signal from the input of the modified acquisition node to the output of the modified mitigation node. Figure 8 shows the configuration for the *FPS* test. Figure 9 shows the diagram. Prior to this test, the control board and test equipment were initialized.

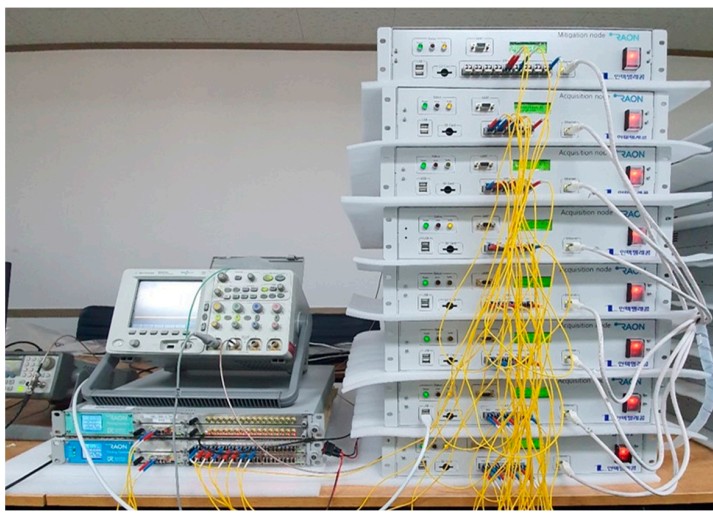

**Figure 8.** Installation of the *FPS*.

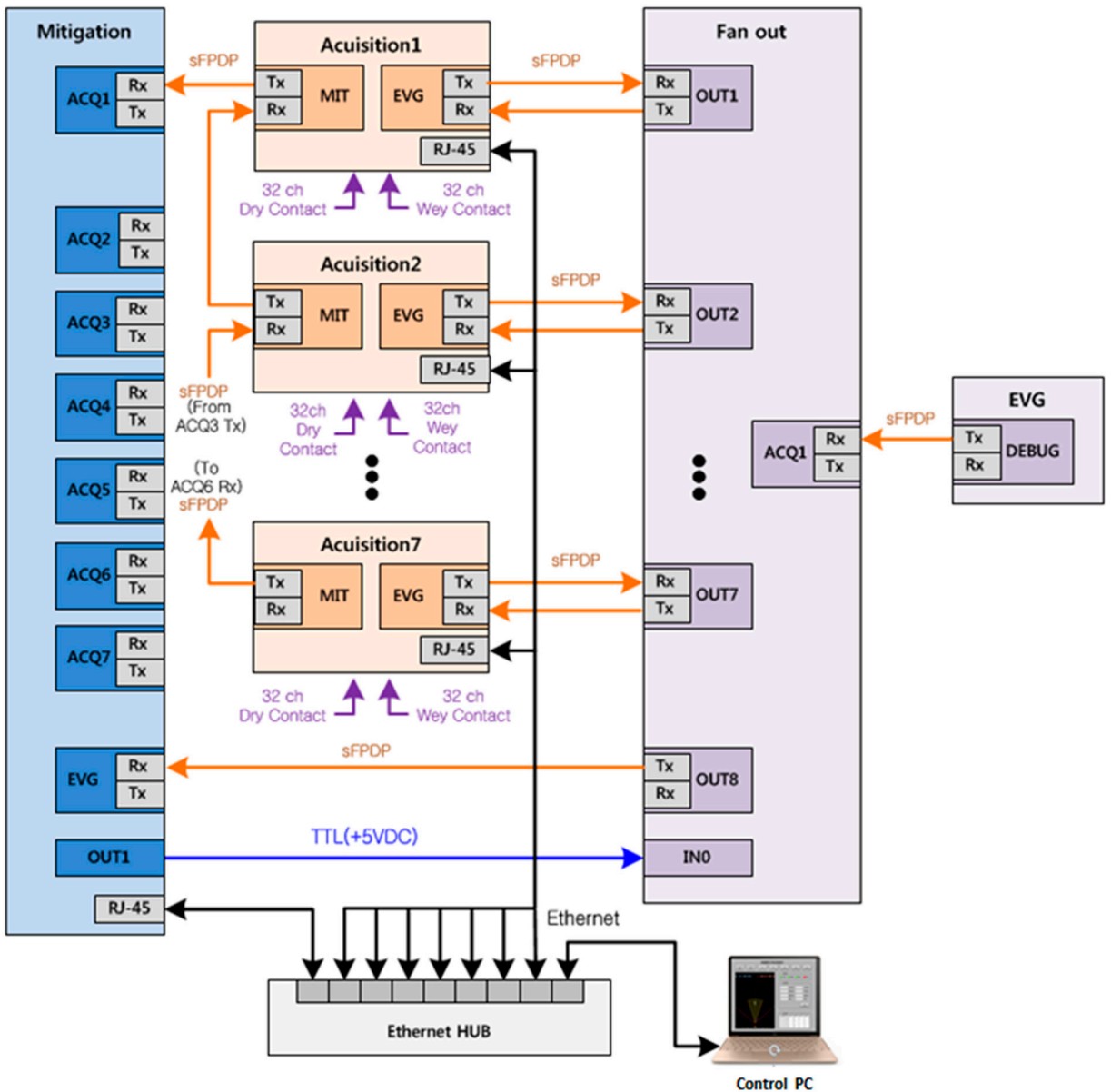

**Figure 9.** Test schematic of the *FPS*.

*5.1. Initial State of the Test Equipment*

The control boards and test equipment must be prepared for initialization and test preparation in the following order.

1.  The power switch of the test equipment must be kept off.
2.  Measures must be taken to prevent static electricity during the test.
3.  It must be ensured that the power cables and other connecting cables of the test equipment are connected properly.
4.  It must be checked that the power required for the test is set to the specified power conditions.
5.  The equipment and all instruments required for the test must be warmed up prior to the test.
6.  A 220 V AC power must be applied to the modified mitigation node, modified acquisition node, *EVG*, waveform generator, Ethernet hub, and oscilloscope.

*5.2. First Test for Communication Check*

The first test examines the cyclic redundancy check (*CRC*) per packet at the end of the packet. The *CRC* shows that the input signal is the same as the output signal. The

test program used Vivado, a Xilinx *FPGA* program. The *CRC* of the data sent to the modified acquisition node is fffffff2, as shown in Figure 10. At this time, the *CRC* data of the received signal of the modified mitigation node are fffffff2, as can be seen in the chip scope screen of Figure 11. It can be found that the input and output signals are the same by checking the input signal of the modified acquisition node as the input signal of the modified mitigation node.

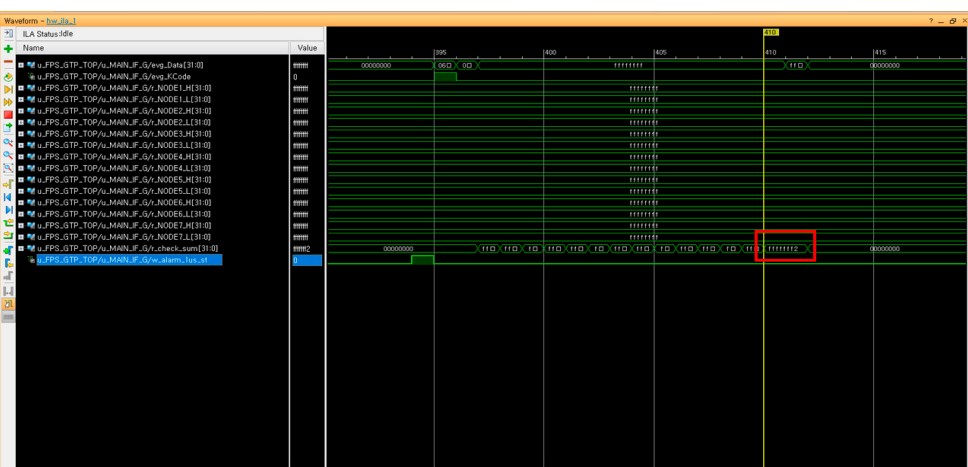

**Figure 10.** CRC data of the packet sent by the modified acquisition node.

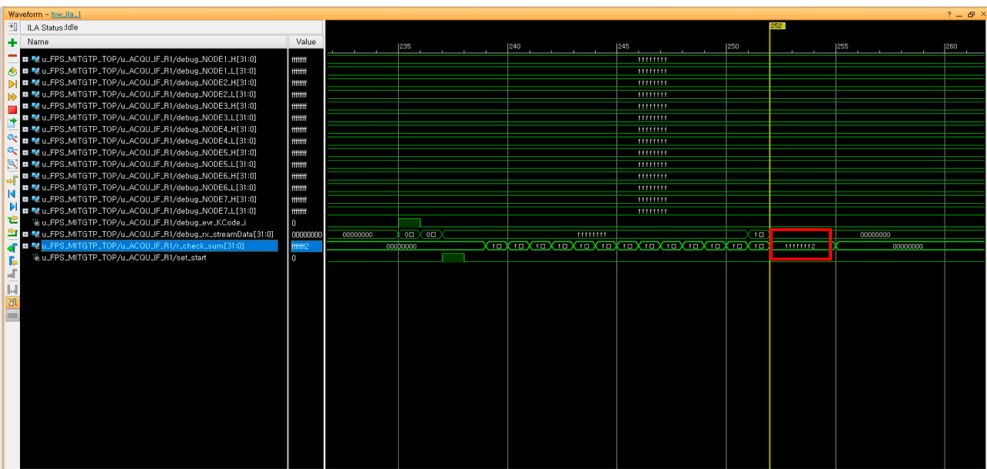

**Figure 11.** *CRC* data of the packed received from the modified mitigation node.

### 5.3. Second Test for Mitigation Processing Time

The second test checks an interlock signal coming into modified acquisition node 7, which is the last node in the chain, sent to the 24 outputs of the modified mitigation node. In this case, instead of using an input signal from the actual diagnostic devices, an input signal from a signal generator that sends 100 ms period pulse and 100 μs pulse width signals was used, as shown in Figure 12. The signal generated from the signal generator is a wet signal of 2 V/0 V. As shown in Figure 13, it can be observed that after the signal generated for an abnormal signal input, the control signal output at the modified mitigation node increases from low (0 V) to high (+5 VDC).

### 5.4. Third Test for FPS Processing Time

The third test is the abnormal signal monitoring and control test in the UI for the operator. The UI was developed with *CSS OPI*. As shown in Figure 14, the UI can check the status of 49 acquisition nodes and one master node on a single screen. It was developed so that detailed node status can be checked by clicking the enter icon next to the node status.

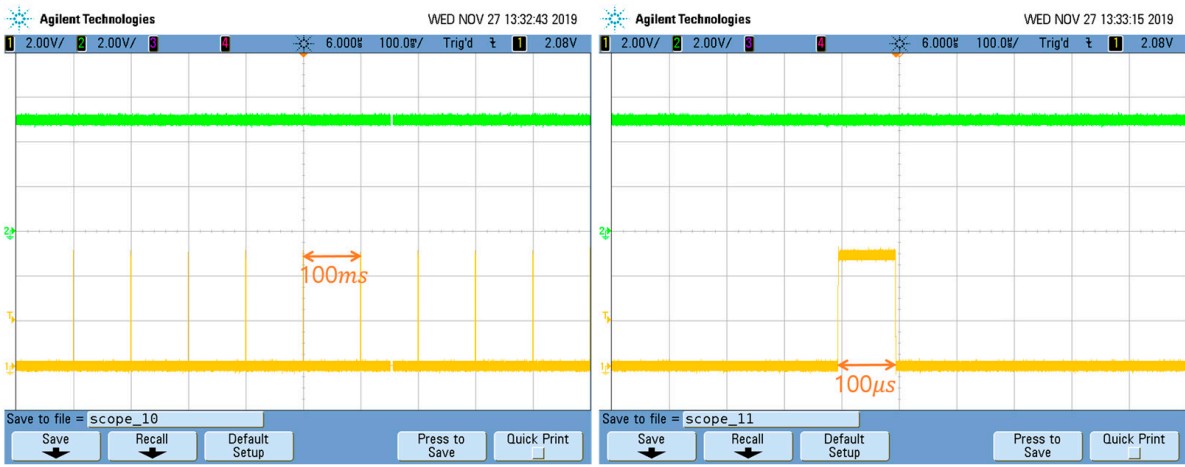

**Figure 12.** Input signal for test (100 ms pulse, pulse width 100 μs).

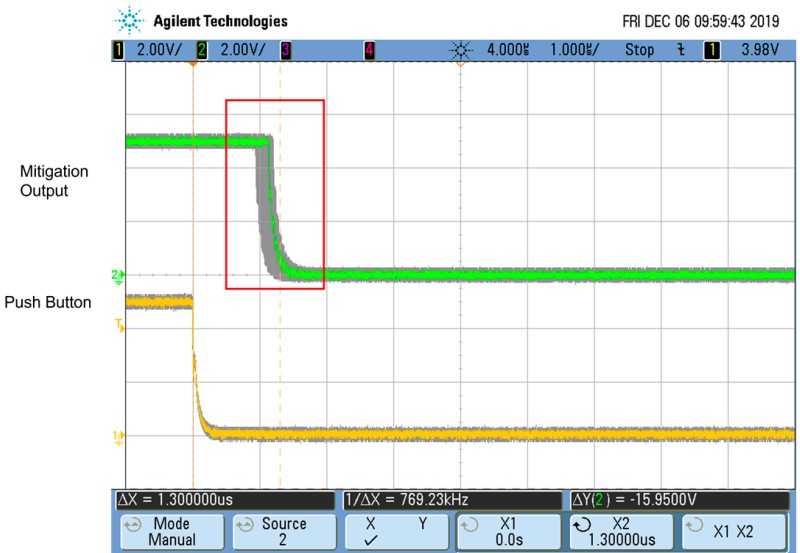

**Figure 13.** Control signal output at the modified mitigation node.

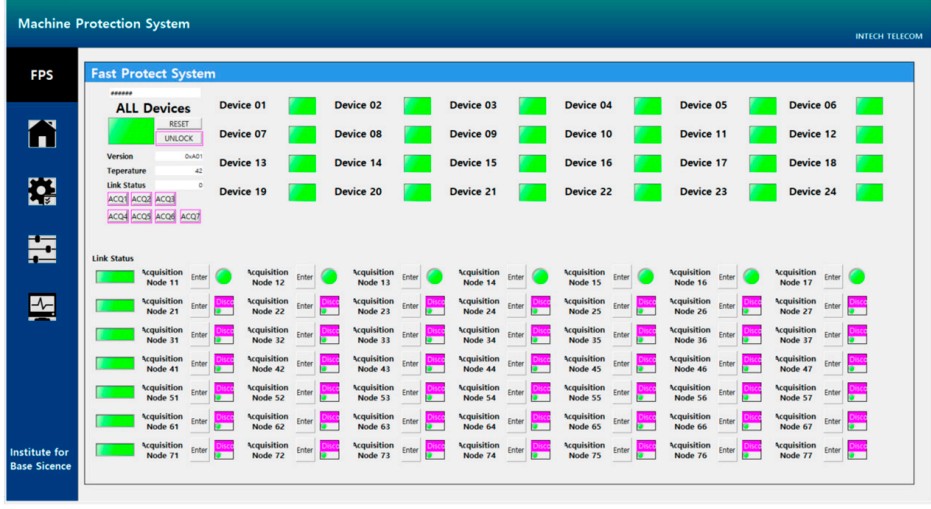

**Figure 14.** Main user interface of fast protection system.

When an abnormal signal enters the acquisition node, the acquisition node at which the abnormal signal entered is displayed in the UI, as shown in Figure 15. If you click enter

next to the node status window with an error, you can check the detailed status of the node, and you can also check the function of saving the *PMS* data at this time to the master node as shown in Figure 16.

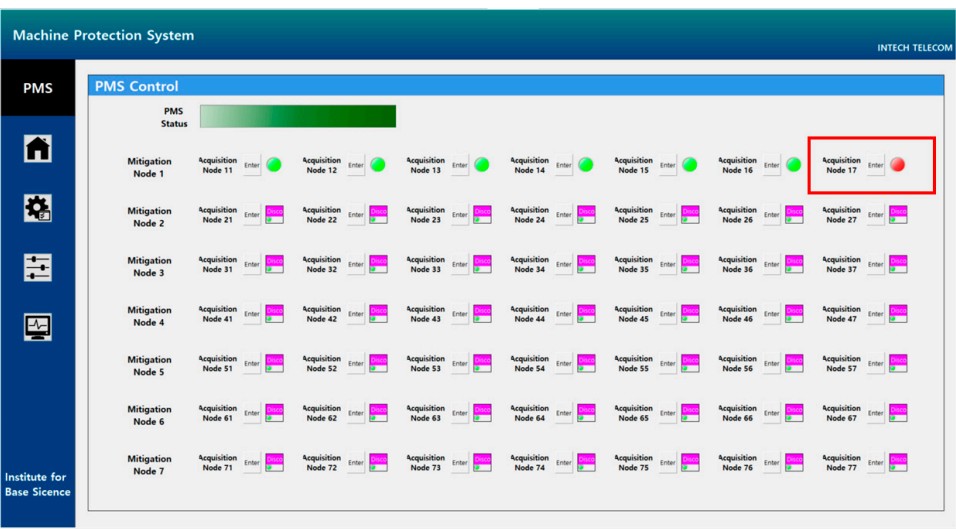

**Figure 15.** Anomaly signal monitoring screen of *FPS* UI.

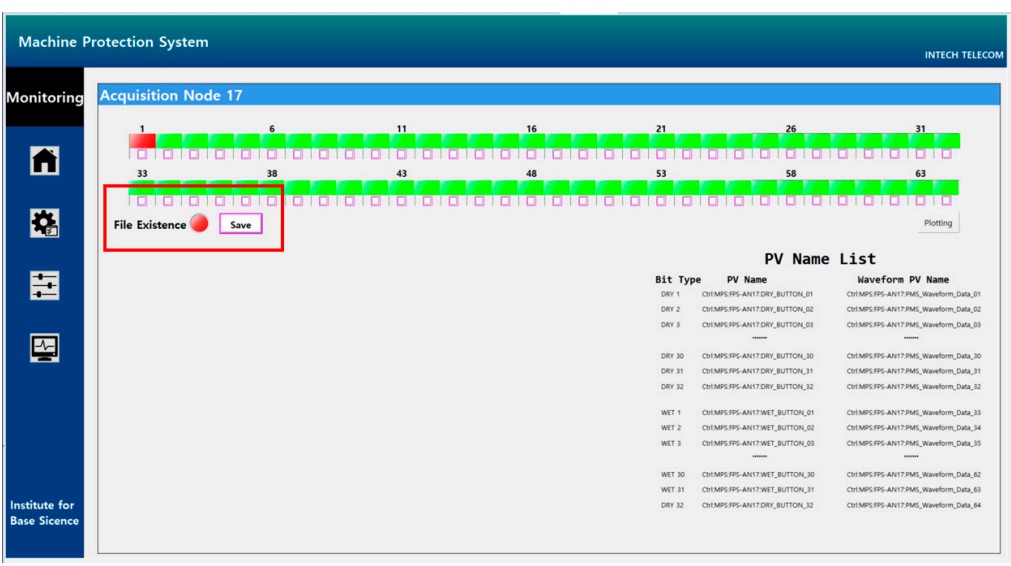

**Figure 16.** Anomaly signal monitoring screen of acquisition UI and PMS data save button.

*5.5. Forth Test for Delay Time in Cable*

The fourth test is one of the most important parts of the *MPS*, which is a delay test from the occurrence of the interlock signal input to the modified acquisition node to the output of 24 control signals connected to the modified mitigation node. The expected delay time for the initial design of optical cable in length of 440 m was 8.3 μs, and it was confirmed that the delay time was within 10 μs.

The delay time for the 1 m optical fiber length is up to 4.92 μs, as shown in Figure 17. The expected delay time of this *FPS* for a 340 m optical cable was 6.52 μs, and it was confirmed that its delay time was not only within 10 μs, but was also shorter than that of the previous prototype version of the *FPS*, which was 7.8 μs for a 340 m optical cable. Considering that the delay time of the optical cable was 5 μs at 1 km, the response time measurement results of the RAON *FPS* met the expected results, which is fast enough to protect the accelerator from beam loss.

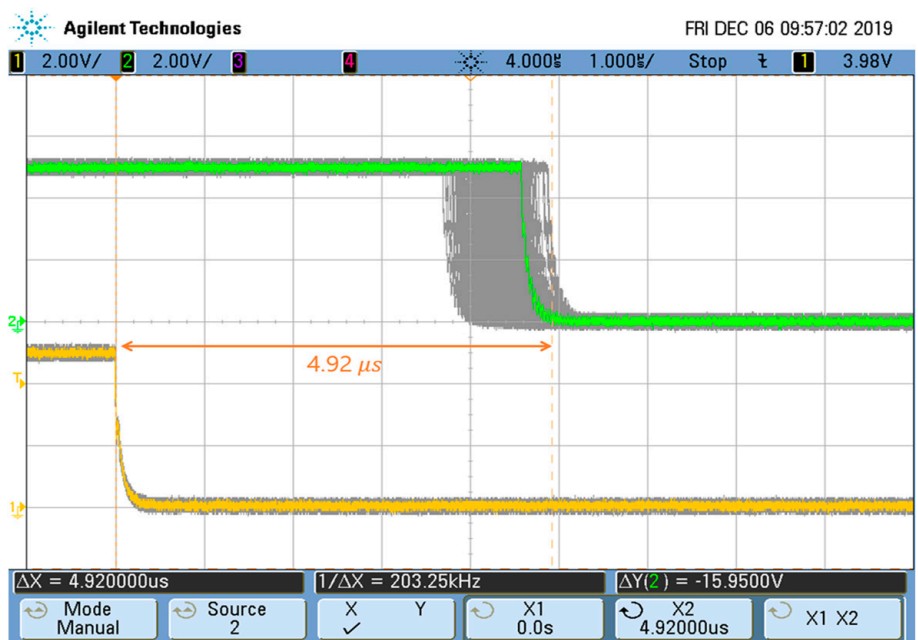

**Figure 17.** Control signal delay time at the modified mitigation node.

The time stamp and *PMS* event codes from external *EVG*s are input into all *FPS* nodes. Whether it can be received through the embedded *EVR* of each node or not was tested. Figure 18 displays the test results. On the chip scope screen of the mitigation and modified acquisition nodes, it can be observed that the year, month, day, hour, minute, second (per sec), second (per μs), and event codes are received normally.

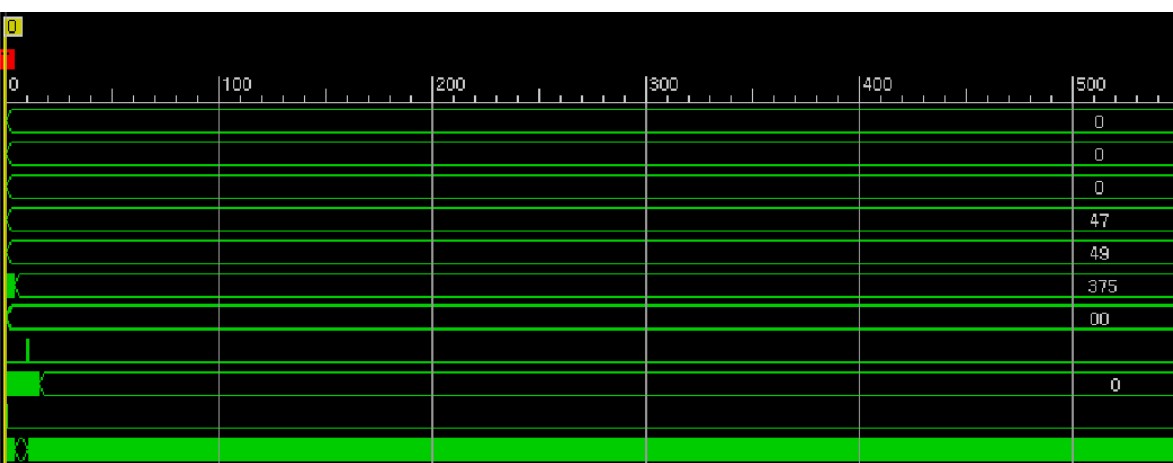

**Figure 18.** Received data in the embedded *EVR*.

An external trigger was generated for the *PMS* test. The trigger stored processing data inside the modified acquisition node of ±200 μs in RAM associated with the ARM processor. The data are saved in the designated location in conjunction with the *EPICS* server. When an interlock occurs, the creation of a *PMS* file is displayed on the modified acquisition node. The interlock data are sent to the modified mitigation node remotely when the save button is pressed. The remotely transmitted data are contained in a .txt file, and 400,000 data are stored at ±200 μs, as shown in Figure 19.

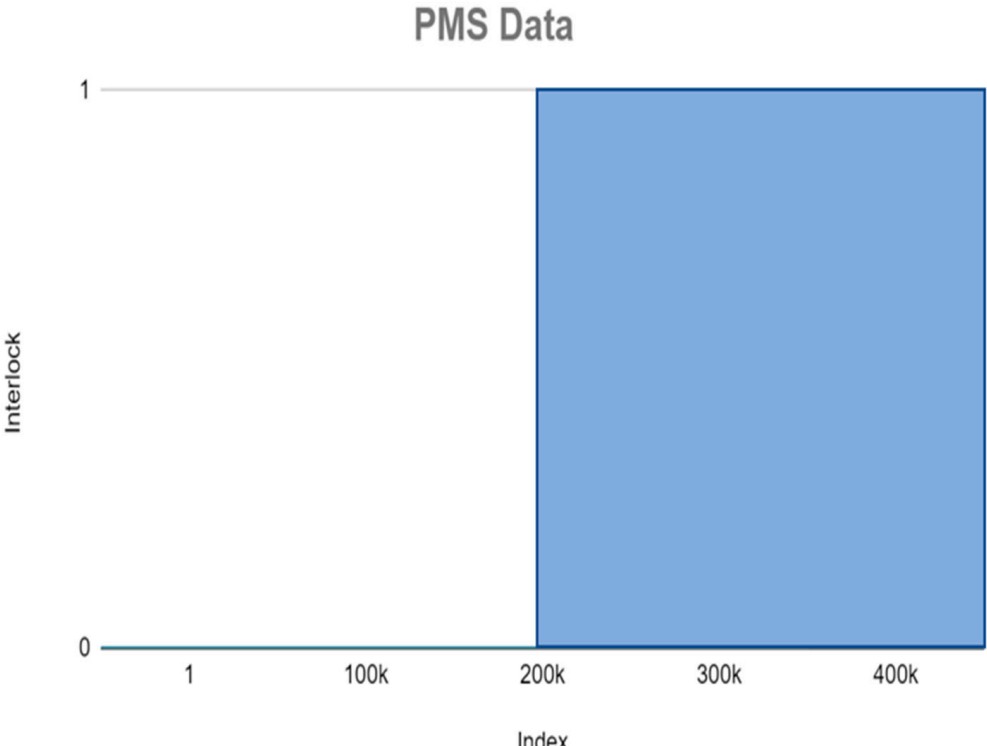

**Figure 19.** Value of index vs. interlock at $\pm 200$ μs.

## 6. Conclusions

The completion of the development process for the Fast Protection System (*FPS*) with Xilinx ZYNQ *SoC* for the RAON heavy-ion accelerator marks a significant milestone. The first *FPS* prototype demonstrated satisfactory performance, meeting the crucial requirement of protecting the internal components of the accelerator from potential beam damage [3]. Notably, the modified mitigation node, which incorporates a cost-effective *SoC*, was successfully optimized by increasing the number of SFP+ ports and implementing a modified driver and library. This enhancement resulted in even faster reaction times compared to the previous version of the *FPS*.

The measured total signal processing time, including measurement and interlock output, for a 1 km optical cable in the RAON system is 9.8 μs, well within the requirement of 20 μs for *FPS* time. The successful achievement of this performance milestone ensures that the *FPS* is capable of effectively responding to potential hazards and mitigating risks within the specified time constraints.

Figure 20 showcases the installation of the actual *FPS*, depicting the practical implementation of the system within the RAON accelerator facility. Looking ahead, future plans involve exploring the integration of artificial intelligence (AI) techniques into the Machine Protection System (*MPS*). By incorporating AI, the aim is to design a system that can predict and proactively control abnormal signals, further enhancing the safety and operational efficiency of the *FPS*.

In summary, the successful development and performance evaluation of the *FPS* with Xilinx ZYNQ SoC for the RAON heavy-ion accelerator highlight the effectiveness and suitability of this integrated solution. The optimized system design, enhanced reaction times, and future plans for AI integration demonstrate the commitment to continuous improvement and innovation in the field of accelerator protection and control systems.

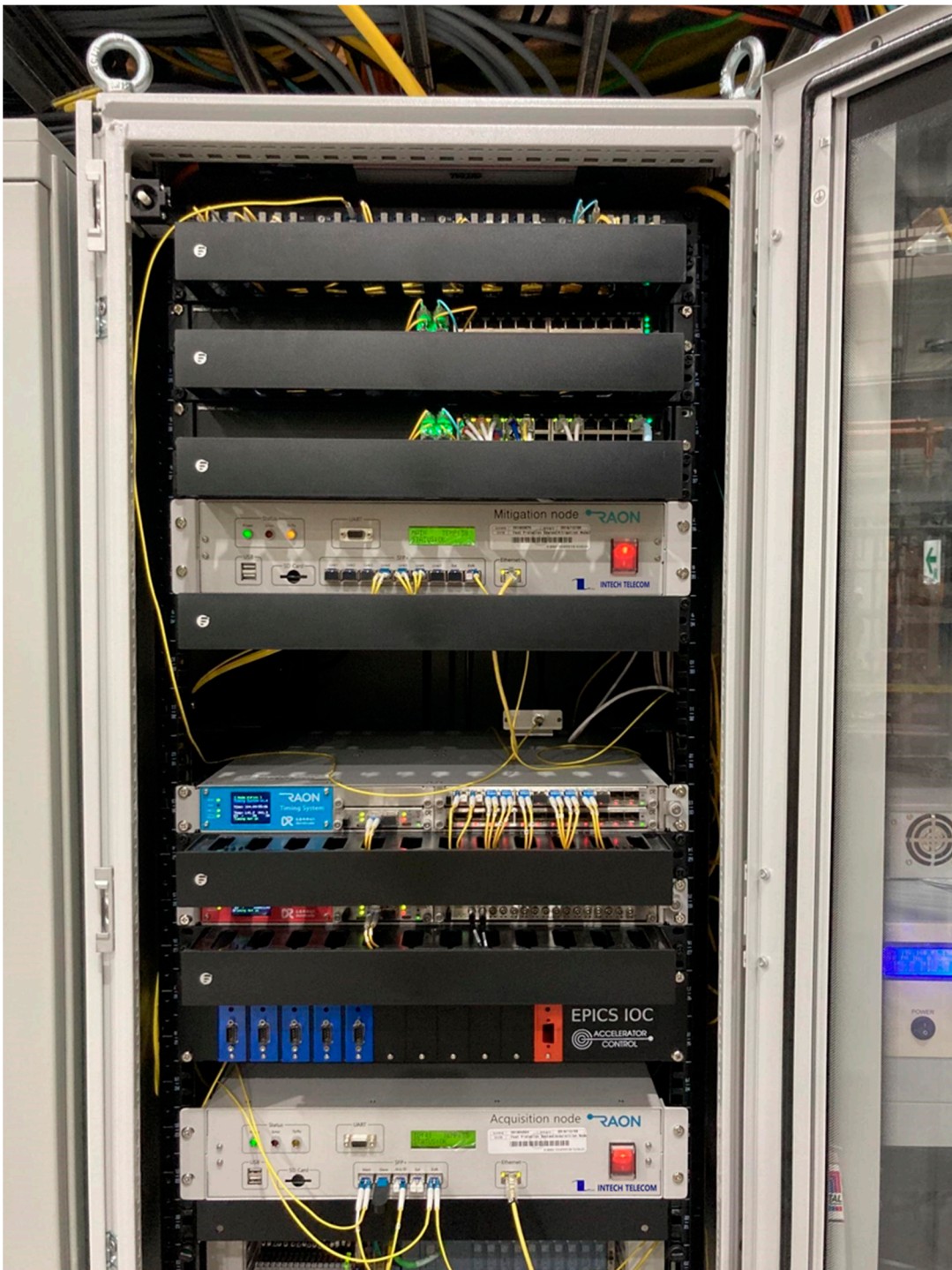

**Figure 20.** Installation of the actual *FPS*.

**Author Contributions:** Conceptualization, S.-H.N.; Methodology, C.S.; Software, S.-H.N.; Validation, C.S.; Formal analysis, S.-H.N.; Resources, C.S.; Data curation, C.S.; Writing—original draft, S.-H.N.; Writing—review & editing, J.B.; Project administration, J.B.; Funding acquisition, J.B. All authors have read and agreed to the published version of the manuscript.

**Funding:** This work was supported by the Rare Isotope Science Project of Institute for Basic Science funded by Ministry of Science, ICT, and Future Planning and National Research Foundation of Korea (2013M7A1A1075764), by the INNOPOLIS Foundation of Korea funded by the Ministry of Science

and ICT (Grant No. 1711149774), and by the National Research Foundation of Korea (NRF) grant funded by the Korea government (MSIT) (No. 2022R1F1A1076188).

**Institutional Review Board Statement:** Not applicable.

**Informed Consent Statement:** Not applicable.

**Data Availability Statement:** Not applicable.

**Conflicts of Interest:** All authors declare no conflict of interest. We have read and agreed to the published version of the manuscript.

## Abbreviation

| Abbreviation | Description |
| --- | --- |
| RISP | Rare Isotope Science Project |
| IBS | Institute for Basic Science |
| SNS | Spallation Neutron Source |
| FRIB | Facility for Rare Isotope Beams |
| SRIM | Stopping and Range of Ions in Matter |
| EPICS | Experimental Physics and Industrial Control System |
| FPGA | Field Programmable Gate Arrays |
| CPU | Central Processing Unit |
| MPS | Machine Protection System |
| SIS | Slow Interlock System |
| FPS | Fast Protection System |
| SoC | System on Chip |
| MPSoC | Multi-Processor System on Chip |
| PS | Processing System |
| PL | Programmable Logic |
| EVG | Event Generator |
| EVR | Event Receiver |
| PMS | Post Mortem System |
| CSS | Control System Studio |
| OPI | Operator Interface |
| IOC | Input Output Controller |
| PV | Processing Value |
| CRC | Cycle Redundancy Check |
| LEBT | Low Energy Beam Transfer |
| RFQ | Radio Frequency Quadrupole |

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
