# Peer review of "Development of Fast Protection System with Xilinx ZYNQ SoC for RAON Heavy-Ion Accelerator"

_applsci, doi:10.3390/app13127121_

Round 1
Reviewer 1 Report
The paper presents the development of an interesting protection system based on Xilinx ZYNQ SoC, applied in the protection of an isotope accelerator facility from beam damage. The topic is interesting, and the quality of the current version of the paper is generally between average and high.
The paper is well written and comprehensive. It seems to make use of appropriate and up-to-date technologies and techniques. The topic seems to be original. It addresses a rare concept in the field, that is the use of FPGA modules to create cheap real-time systems that fulfills specific timing constraints (in this case a control response under 50μs). The conclusion addresses sufficiently the main research question posed and are consistent with the evidence and arguments presented. The references are sufficient and appropriate.
Overall the paper seems to be of sufficiently good quality.
I have no particular comments, beyond that perhaps a Results discussion section (or as a subsection within section 6. Test of Fps) could provide a useful synopsis of the results obtained during the tests against the specifications set to be obtained.
English language in general is fine.
Author Response
Dear Reviewer,
Thank you for your valuable feedback on our manuscript 'Development of Fast Protection System with Xilinx ZYNQ SoC for RAON Heavy-Ion Accelerator'. We appreciate your time and effort in reviewing our work. We have carefully considered your comments and suggestions, and we would like to respond as follows:
- I have no particular comments, beyond that perhaps a Results discussion section (or as a subsection within section 6. Test of Fps) could provide a useful synopsis of the results obtained during the tests against the specifications set to be obtained.
- Thank you for your comment. The FPS test result as well as Fig. 14~16 is improved in section 5 (test result) to describe explanation of developed FPS system performance and 6 (conclusion).
We believe these revisions have significantly improved the quality and clarity of our work.
Once again, we sincerely appreciate your time and expertise in reviewing our manuscript.
Thank you for considering our rebuttal.
Sincerely,
Jungbae Bahng

Reviewer 2 Report
- language needs to be revised. so words used are confusing in meaning. revise
- No reference should be put in the Abstract.
- The first sentence does not provide any information. Be direct about what is or was done.
- Abstract should address the problem and how the research development served to solve it.
- I suggest a section for Abbreviations to help readers as there are a number of them.
- the problem concerning the FPS should be clarified if it going to be addressed in the introduction. The descriptive introduction is not informative about where the actual problem lies and is better defined.
- in Section 3: the first mention of the problem is made - a little late.
- line 85: where is the sentence?
- At some points, there is no clear definition between the actual system implemented and the required modifications.
- Need to see an application of this system using acquired real parameters of the reactor system. In other words, there are conditions stated about the failure of the FPS system. Before proceeding, the previously compiled data in case of near failure can be used to simulate a pseudo-real situation and be inputted into the system to see hypothetically how it may interact with a potentially critical situation that may be faced. This must be done before implementing anything in the accelerator. Isn't that the purpose of the study?
at this point, a conclusion may be made.
-
- language and words are narrative and confusing in most parts of the manuscript
Author Response
Dear Reviewer,
Thank you for your valuable feedback on our manuscript 'Development of Fast Protection System with Xilinx ZYNQ SoC for RAON Heavy-Ion Accelerator'. We appreciate your time and effort in reviewing our work. We have carefully considered your comments and suggestions, and we would like to respond as follows:
1. Language needs to be revised. so words used are confusing in meaning. revise
- The manuscript had been revised by English editing service.
- No reference should be put in the Abstract.
- The position of reference is moved in the text section from the abstract.
- The first sentence does not provide any information. Be direct about what is or was done.
- We revised as your suggestion.
- Abstract should address the problem and how the research development served to solve it.
- Thank you for your comment. Abstract has been re-written to include the problem, method and result.
- I suggest a section for Abbreviations to help readers as there are a number of them.
- Thank you for your comment. List of abbreviations is updated in Table 2.
- The problem concerning the FPS should be clarified if it going to be addressed in the introduction. The descriptive introduction is not informative about where the actual problem lies and is better defined.
- Thank you for your feedback and suggestions for improving the introduction section. We have carefully considered your comment and revised enhance the clarity and informativeness of the introduction. The updated introduction now provides a more comprehensive overview of the Rare Isotope Science Project (RISP) and its accelerator facility, highlighting the importance of the Machine Protection System (MPS) in safeguarding the internal components of the accelerator from beam damage.
- in Section 3: the first mention of the problem is made - a little late.
- Thank you for your comment. A part of section 3 for research motivation is re-positioned in section 1 (Introduction).
- line 85: where is the sentence?
- We revised it.
- At some points, there is no clear definition between the actual system implemented and the required modifications.
à Thank you for pointing out this important aspect. The modified version focuses on reducing latency for quick shut off in the FPS. The modifications are described on both the mitigation node and the acquisition node. The changes in hardware specifications, such as the addition of SFP+ ports, the replacement of certain components, and the inclusion of new features, are clearly outlined for each node.
- Need to see an application of this system using acquired real parameters of the reactor system. In other words, there are conditions stated about the failure of the FPS system. Before proceeding, the previously compiled data in case of near failure can be used to simulate a pseudo-real situation and be inputted into the system to see hypothetically how it may interact with a potentially critical situation that may be faced. This must be done before implementing anything in the accelerator. Isn't that the purpose of the study? at this point, a conclusion may be made.
à In our study, we conducted simulations to determine the stopping power and melting time of the accelerator components when subjected to ion beam collisions. These simulations provided valuable data that guided the development of the Fast Protection System (FPS). Based on the simulation results, we designed and implemented the FPS system, ensuring its compatibility with the accelerator's requirements. During the testing phase, we evaluated the performance of the FPS system and compared it to the specified requirement of 20 us. The results were promising, as the FPS system achieved a reasonable response time that satisfied the specified requirement.
We believe these revisions have significantly improved the quality and clarity of our work.
Once again, we sincerely appreciate your time and expertise in reviewing our manuscript.
Thank you for considering our rebuttal.
Sincerely,
Jungbae Bahng

Reviewer 3 Report
The content studied in the article is interesting, but still requires a major revision before publication. The detailed comments are as follows:
1. All the images in the manuscript have poor quality and are blurry, requiring improvement
2. The author should provide a detailed description of the innovative points of the manuscript, as well as the actual work and technical indicators, in the abstract section
3. The author should provide a detailed research background in the introduction section, as well as a review and comparison with existing related technologies
4. English grammar writing is very poor. Professional polish is recommended
The content studied in the article is interesting, but still requires a major revision before publication. The detailed comments are as follows:
1. All the images in the manuscript have poor quality and are blurry, requiring improvement
2. The author should provide a detailed description of the innovative points of the manuscript, as well as the actual work and technical indicators, in the abstract section
3. The author should provide a detailed research background in the introduction section, as well as a review and comparison with existing related technologies
4. English grammar writing is very poor. Professional polish is recommended
Author Response
Dear Reviewer,
Thank you for your valuable feedback on our manuscript 'Development of Fast Protection System with Xilinx ZYNQ SoC for RAON Heavy-Ion Accelerator'. We appreciate your time and effort in reviewing our work. We have carefully considered your comments and suggestions, and we would like to respond as follows:
- All the images in the manuscript have poor quality and are blurry, requiring improvement
- Thank you for your comment. All the images are updated.
- The author should provide a detailed description of the innovative points of the manuscript, as well as the actual work and technical indicators, in the abstract section
- Thank you for your comment. Abstract has been re-written to include the problem, method and result.
- The author should provide a detailed research background in the introduction section, as well as a review and comparison with existing related technologies
- Thank you for your feedback and suggestions for improving the introduction section. We have carefully considered your comment and revised enhance the clarity and informativeness of the introduction. The updated introduction now provides a more comprehensive overview of the Rare Isotope Science Project (RISP) and its accelerator facility, highlighting the importance of the Machine Protection System (MPS) in safeguarding the internal components of the accelerator from beam damage.
- English grammar writing is very poor. Professional polish is recommended
- The manuscript had been revised by English editing service.
We believe these revisions have significantly improved the quality and clarity of our work.
Once again, we sincerely appreciate your time and expertise in reviewing our manuscript.
Thank you for considering our rebuttal.
Sincerely,
Jungbae Bahng

Round 2
Reviewer 2 Report
- the text has improved considerably
- the additional information and data provided clarified the objective of the study and the results
Reviewer 3 Report
The author has made the necessary modifications as requested, and we recommend that it be accepted.